# Diet and Blood Concentrations of Essential and Non-Essential Elements among Rural Residents in Arctic Russia

**DOI:** 10.3390/nu14235005

**Published:** 2022-11-24

**Authors:** Tatiana Sorokina, Nikita Sobolev, Natalia Belova, Andrey Aksenov, Dmitriy Kotsur, Anna Trofimova, Yulia Varakina, Andrej M. Grjibovski, Valerii Chashchin, Rimma Korobitsyna, Yngvar Thomassen

**Affiliations:** 1Arctic Biomonitoring Laboratory, Northern (Arctic) Federal University Named after M. V. Lomonosov, Naberezhnaya Severnoy Dvini 17, 163002 Arkhangelsk, Russia; 2Central Scientific Research Laboratory, Northern State Medical University, Troitskiy Ave., 51, 163000 Arkhangelsk, Russia; 3N. Laverov Federal Center for Integrated Arctic Research, Ural Branch of the Russian Academy of Sciences, Naberezhnaya Severnoy Dvini 23, 163000 Arkhangelsk, Russia; 4Department of Epidemiology and Modern Vaccination Technologies, Sechenov First Moscow State Medical University (Sechenov University), 119991 Moscow, Russia; 5Department of Health Policy and Management, Al-Farabi Kazakh National University, Almaty 050040, Kazakhstan; 6West Kazakhstan Marat Ospanov Medical University, Aktobe 030019, Kazakhstan; 7Research Laboratory of Complex Problems of Hygiene and Epidemiology, North-Western State Medical University Named after I.I. Mechnikov, Kirochnaya ul. 41, 191015 Saint-Petersburg, Russia; 8Institute of Ecology, National Research University Higher School of Economics, Myasnitskaya Str. 20, 101000 Moscow, Russia

**Keywords:** country foods, dietary intake, trace metals, nutrients, iron status, indigenous population

## Abstract

Nutrition is an essential factor for human health. Earlier research has suggested that Arctic residents are vulnerable to environmental toxic exposures through traditional foods. Although Russia is the largest Arctic country, the evidence on the topic from the Russian part of the Arctic is scarce. We studied associations between blood concentrations of essential and non-essential elements and traditional food consumption in 297 adults from seven rural settlements in the Nenets Autonomous Area, Northwest Russia. Blood arsenic concentration was positively associated with consumption of rainbow smelt, pink salmon, Arctic char and navaga fish. Frequent consumption of northern pike was associated with increased concentration of blood mercury. Blood mercury and arsenic concentrations were significantly associated with blood selenium. We also observed positive associations between blood lead levels and the frequency of goose consumption. Moreover, subjects who reported to be hunters had higher blood levels of lead, suggesting contamination of goose meat with fragments of shotgun shells. Blood cobalt and manganese concentrations were inversely associated with serum ferritin levels. Positive associations between blood levels of manganese and lead were observed. Moreover, blood lead concentrations were significantly associated with cadmium, mercury, copper, and zinc. Our results corroborate earlier findings on the traditional foods as source of non-essential elements for the Arctic residents. Observed correlations between the levels of lead and other elements warrant further research and may have potential implications for the studies on the associations between essential elements and health outcomes.

## 1. Introduction

Among Arctic indigenous peoples’ traditional foods are an essential part of their nutrition with hunting and fishing as important activities to ensure provision of food and other needs of the people. Traditional foods are not only major sources of essential micronutrients but are also potential sources of persistent toxic inorganic pollutants (PIPs) [1]. Intake of sea mammals, fish and other seafood have consistently been shown as main sources of mercury (Hg), lead (Pb), cadmium (Cd) and arsenic (As) for Arctic residents [1,2,3,4]. These foods contribute also significantly to the intake of essential micronutrients such as copper (Cu), cobalt (Co), iodine (I), selenium (Se) and zinc (Zn) [4,5]. Reindeer meat is also a main food item and a source of micronutrients for Arctic residents. In a Norwegian study, this meat has been shown to have more iron (Fe), Zn and Se compared with other meats such as beef, lamb, pork, and chicken, but low amounts of As, Cd and Pb were also found [6].

Nenets Autonomous Okrug (NAO) is a region located in the north-eastern part of European Russia and is bounded by the Kara, the Barents, and the White seas. The Nenets (7504 people in 2020), also known as Samoyed, is a Samoyedic ethnic group native to northern Arctic Russia constituting 18% of the total population of NAO [7]. Significant changes in way of life have occurred over the last two decades among the Nenets in NAO and the neighbouring Yamal-Nenets Autonomous Okrug, distancing them from their ancestors in economic, social, cultural, and anthropometrical respects [8,9]. Consumption of traditional foods is now at its highest during the seasons of fishing and reindeer slaughter, while out of these seasons, the consumption of easily digestible carbohydrates has increased. Seasonal fishing and traditional migration routes for reindeer husbandry are being disturbed due to climate change, and this has also decreased the consumption of traditional foods [10]. In addition, it is possible that change in diet results from reforms of the Soviet period or from shifts in nutrition during the last two decades [11].

A Russian mega-grant study was used to establish an accredited analytical bio-monitoring laboratory for quantifying the presence of persistent organic pollutants (POPs), PIPs and micronutrients in food items consumed by pan-Russian Arctic indigenous populations [12]. This initiative has contributed to the improvement of the Russian system of biomonitoring in general and laboratory capacity in particular [1].

We have recently reported concentrations of selected essential and non-essential elements in fish consumed and in whole blood among Nenets and non-Nenets residents of NAO [4,12]. The whole blood concentrations of manganese (Mn), Zn, Pb and As in the coastal sub-group were significantly higher while Cu and Hg were higher in the inland population. Additionally, being Nenets or non-Nenets, gender, smoking and to lesser extent, age, and body mass index (BMI), had impact on the whole blood concentrations. The high prevalence of Fe-deficiency (S-ferritin < 15 μg/L) among Nenets revealed in that study, especially among menstruating women (26.6%), women (≤50 years), calls for further attention when evaluating nutritional intake and trace element status in a population since there are close relations between metabolism of some elements and Fe status. Increased expression of common intestinal Fe and other divalent element transporters (e.g., divalent metal ion transporter 1 (DMT 1)) induced by Fe deficiency influence metabolic imbalances and changes in the trace element status, particularly Co and Mn, but Pb and Cd have also been implicated [13,14,15,16,17].

In the current work, the reported dietary intake of traditional food items among Nenets and non-Nenets living in seven different rural settlements of NAO is evaluated, and associations are examined between dietary determinants, lifestyle habits, Fe status and concentrations of essential and non-essential elements measured in whole blood.

## 2. Materials and Methods

The site description, study design, background data of the participants and their whole blood concentrations of essential and non-essential elements have been described elsewhere [12]. Briefly, the sample consisted of 204 women and 93 men aged 18–87 years, recruited from seven rural settlements in NAO, with Nenets (54.2%) and Russians (35.7%) as the main population groups. Of these, 148 and 149 lived in coastal and inland settlements, respectively.

### 2.1. Implementation

The fieldwork was conducted from July to October 2018. The residents of the settlements were invited to participate in the study through information campaigns in the villages. According to the study, protocol participants were recruited based on only two criteria: (i) age 18 years (the age of maturity under Russian law) or older, and (ii) permanent residence in the settlement. There were no specific requirements for gender, nationality, profession, lifestyle and nutrition. Non-fasting whole blood was collected during day-time examination at the medical units of the settlements. The participants responded to a self-administrated food frequency questionnaire (FFQ) to record nutritional variables under the guidance of trained medical personnel who checked the completeness of the questionnaires. The study protocol was approved by the Ethical Committee at the Northern Medical State University, Arkhangelsk (no. 06/09-17). All participants gave their written informed consent following the Helsinki Declaration.

### 2.2. Food Frequency Questionnaire

The FFQ based on AMAP reports [18] was specifically developed for this study. Further information is from Sobolev et al., 2019 [19]. The participants completed a semi-quantitative FFQ that measured their previous monthly prevalence of consumption of nutritional items of five food categories: meat; birds; fish; berries and mushrooms using a 6-point scale from daily consumption, one to three times per week, three times a month, once a month, less to “one per month” or never. For each product, participants themselves wrote down the approximate amounts in grams of one serving. This information, however, was not used, since the information exhibited significantly greater inter-individual sources of error than the information on the frequency of consumption. The fish species consumed are grouped into anadromous (Atlantic salmon, pink salmon, chum salmon, Arctic char, Arctic cisco, rainbow smelt), semi-anadromous (broad whitefish, humpback whitefish), marine (navaga, plaice, cod, herring) and freshwater (burbot, northern pike, other local fishes) fishes (Table 1).

### 2.3. Measurements of Elements in Whole Blood and Ferritin in Serum

The collection procedures for biological samples and procedures for the measurements of elements and serum ferritin (S-ferritin) have been described elsewhere [12]. Briefly, whole blood was collected from the cubital vein with either vacutainer tubes having EDTA for element measurements or vacutainer tubes with clot activator for serum preparation. All whole blood and serum samples were immediately frozen before transportation to Northern Arctic Federal University (NArFU) for storage at −25 °C until analysis. Whole blood was analysed for the “total” content of elements by inductively coupled argon plasma mass spectrometry after dissolution in nitric acid. No speciation methods were available for As, Hg and Se. Serum ferritin was measured by a Random Access automatic biochemical analyser using ferritin assay kits (Biosystem, Barcelona, Spain).

The measurements showed that the concentrations of several essential and non-essential elements varied significantly in whole blood according to gender, population group and locality. Cd levels among non-Nenets non-smokers (0.19 μg/L) indicated a dietary intake at a natural global background level. Hg in whole blood showed that 10% of women in the fertile age had a Hg intake above the European Food Safety Authority’s recommendation. The Pb concentrations were in the range of, or partly exceeding reference values for increased risk of nephrotoxicity. A high prevalence of Fe deficiency among menstruating women (<50 years) (26.6%) was also identified.

### 2.4. Statistics

The distributions of continuous variables were visually assessed, and the skewness of the distributions were calculated. When the skewness of a distribution exceeded 2.0, the variables were log-transformed. We also accessed normal distribution by Q-Q plots and therefore B-As, B-Cd, B-Co and B-Pb have been log-transformed. The arithmetic means (AM) were calculated for variables with a normal distribution. Geometric means (GM) are presented for non-normally distributed variables.

Values below detection limit (DL) were substituted with ½ DL. Students *t*-test was used for the assessment of differences in continuous variables between two groups. General linear models were used when participants were stratified into subgroups for the purpose of visualizing dose–response associations adjusted for relevant variables. The Chi-squared test was used for assessing group differences in categorical variables. Least square regression analysis was used to assess univariate associations with Pearson’s correlation coefficient (Pearson’s r) as the measure of association. An initial multiple linear regression analysis (backward procedure) was used to assess the whole blood concentrations as dependent variables of the various elements, considering potential impact of age, S-ferritin, gender, current smoking, BMI and B-Pb concentrations. Each food item (see Table 1) was entered as a categorical variable into the model that occurred after the regression analysis. The regression models were assessed for multicollinearity by calculating the variance inflation factor and tolerance. Two-tailed *p*-values < 0.05 were considered as the level of statistical significance. The statistical package SPSS^®®^, version 25.0 (IBM Corp., Armunk, NY, USA) was used for the calculations.

## 3. Results

The prevalence of subjects reporting to consume local fish, meat, birds, berries, or mushroom and the previously published whole blood concentrations of elements and S-ferritin among the participants are presented in Table 1. The prevalence of subjects consuming anadromous and marine fish, reindeer meat, goose and cloud-/cowberries were high. Significantly, more non-Nenets consumed anadromous, semi-anadromous and freshwater fish in addition to reindeer compared to the Nenets. Even though more non-Nenets than Nenets consumed reindeer meat, the median number of meals consumed each month was similar (not tabulated). Nenets also consumed significantly more meals with goose meat each month. For other food items, no significant difference was found between the prevalence and number of meals per month.

More subjects from coastal populations consumed goose, marine fish, and mushroom, while consumption of semi-anadromous and freshwater fish, blueberries and cowberries was more often reported among inland inhabitants. There were marginal differences in the percentage of men and women consuming different traditional foods, but men consumed more semi-anadromous fish, while women consumed more cloud- and cowberries.

Appendix A shows the whole blood elements concentrations that differed at a *p* < 0.10 level between participants consuming a particular food item compared to those not consuming the food item. These food items were included in the multiple linear regression models for the respective elements considering age, BMI, S-ferritin, B-Pb (log), current smoking, and gender. B-Pb was included due to the high number of significant associations with other elements (Table 2) at low B-Pb concentrations and the well-known impact of Pb on the erythropoiesis at equivalent B-Pb concentrations.

Both B-Mn and B-Co are decreasing by increasing S-ferritin. Age, gender, and consumption of reindeer meat are also associated with B-Co. To study the relationship between concentration of S-ferritin and increase in B-Mn and B-Co, the subjects were stratified into six subgroups.

Figure 1 shows that the increase in B-Mn starts below S-ferritin concentrations of 50 μg/L among all participants and below 30 μg/L among females. An increase in B-Co concentrations starts at S-ferritin concentrations <30 μg/L (Figure 2).

B-Mn was also associated with the concentrations of B-Pb. When stratifying the participants into tertiles with low, medium, and high S-ferritin and low, medium, and high B-Pb, it can be shown that B-Mn decreases with higher S-ferritin and increases when B-Pb increases within each stratum of B-Pb (Figure 3).

Consumption of mushroom, other local meat than reindeer, BMI, B-Pb, and S-ferritin are all associated with the B-Se concentrations, but none of the groups of fishes were associated with B-Se. However, to assess fish as a potential source of Se, B-Hg and B-As were added to the model. This analysis shows that B-Se was associated with B-Hg and lgB-As, while the other variables were not required in the model (B-Se = 111 *^p^*
^< 0.001^ + 1.44 B-Hg *^p^*
^< 0.001^ +9.71 lgB-As *^p^*
^< 0.001^).

When stratifying B-As and B-Hg into tertiles (Figure 4), the B-Se concentrations clearly increase with increasing amounts of As and Hg in the whole blood. Subjects in the highest tertile of both B-As and B-Hg had AM B-Se of 143 μg/L compared with 111 μg/L among those in the lowest tertile of B-As and B-Hg.

Consumption of marine fish is significantly associated with higher B-As. Additionally, the consumption of anadromous fish was associated with B-As. However, when both types of fish were introduced into the same regression model, anadromous fish consumption did not achieve statistical significance (*p* = 0.08). All single fish species belonging to the group of marine fish or anadromous fish were assessed in a multiple linear regression.

Subjects reporting to eat navaga, rainbow smelt, pink salmon and Arctic char had higher B-As. When these four species were introduced simultaneously into the model, the following relationship was calculated: B-As (lg) = −0.38 *^p^*
^= 0.02^ + 0.20 *^p^*
^= 0.02^ B-Pb (lg) + 0.009 *^p^*
^< 0.001^ Age + 0.18 *^p^*
^= 0.01^ navaga + 0.25 *^p^*
^= 0.001^ rainbow smelt + 0.20 *^p^*
^= 0.006^ pink salmon + 0.14 *^p^*
^= 0.04^ Arctic char. Figure 5 shows the B-As concentrations according to the number of the above-mentioned fish species consumed by the participants adjusted for B-Pb and age.

Freshwater fish consumption was positively associated with the B-Hg concentrations (Table 3). Further detailed regression analysis showed northern pike as the only fresh-water fish associated with B-Hg concentrations (results not shown). Subjects reporting to eat northern pike (*n* = 143) had B-Hg concentration of 6.6 μg/L vs. 4.0 μg/L among those not eating northern pike (*n* = 154) (*p* < 0.001).

The results from the multiple linear regression analysis show that men and current smokers have higher B-Pb than women and non-smokers (Table 3). The consumption of goose is also associated with higher B-Pb concentrations. Forty participants did not respond to the question of being a hunter. Thus, we have chosen to evaluate the impact of goose consumption on B-Pb concentrations among all participants and those who reported to be non-hunters (*n* = 202). When adding the number of meals with goose meat each month instead of eating goose or not to the regression model shown in Table 3, the following regression model relationship could be calculated among all participants: B-Pb (lg) = 1.24 *p* < 0.001 + 0.12 *p* = 0.003 current smoking + 0.19 *p* < 0.001 gender + 0.01 *p* < 0.001 number of goose meals. “Number of meals with goose meat” was also of statistical significance when only non-hunters were included in the model.

Figure 6 shows that B-Pb increases with number of meals with goose meat per month in both groups, but more among all participants. Subjects reporting to be hunters have higher GM (and 95% CI) B-Pb concentrations than subjects reporting not to be a hunter after adjusting for gender, current smoking, and number of meals with goose meat each month (32.7 μg/L; 26.1–40.7 vs. 21.9 μg/L; 19.7–24.4, *p* = 0.003) (not tabulated).

The concentrations of B-Cd, B-Cu and B-Zn were not associated with any of the recorded food items. B-Pb was positively associated with B-Zn and B-Cd, while S-ferritin was not associated with any of the three elements (results not shown).

A negative association between B-Cd [15,20] and B-Pb [14] with S-ferritin in children and pregnant and non-pregnant women has previously been reported. Among non-smokers in the present study, however, there is absolutely no association between B-Cd and S-ferritin (r = 0.01, *p* = 0.99). We also did not find any association between S-ferritin and B-Pb. B-Cd was higher in the smokers [12], but not associated with any food items.

## 4. Discussion

### 4.1. Traditional Food Consumption

There is a slight difference between the percentage of Nenets and non-Nenets reporting consumption of traditional foods. Reporting consumption of a food item does not infer the actual amount of food consumed. Estimating portion sizes of foods, both when examining displayed foods and when responding to an FFQ about foods previously consumed, cannot be used to calculated absolute intake [21]. While the differences in the food item consumption between men and women and Nenets and non-Nenets are marginal, the place of living has a substantially larger impact. Marine fish and goose are more easily available in coastal villages, while semi-anadromous and freshwater fish and berries are more obtainable in inland settlements. This is mirrored in the significant differences in the prevalence of consumption of these traditional foods between the coastal and inland populations. This may explain the geographical variations in whole blood concentrations of several essential and non-essential elements that we have previously described [12].

### 4.2. Traditional Food Consumption and Elements in Whole Blood

Although several differences in the elemental whole blood concentrations were observed between consumers and non-consumers of different traditional food items (Appendix A), most of them disappear in the multiple linear regression analysis when S-ferritin, sex, B-Pb (lg), current smoking, age, and BMI are considered. The significantly higher whole blood concentrations of As in the coastal population, as earlier described, may be associated with consumption of marine and anadromous fish. The association with marine fish is not unexpected, since it is well known that these fish species are rich in non-toxic organo-arsenic compounds. When the fish species contained in the marine fish group are considered separately, navaga is the species contributing most to the B-As concentrations. Reports of consumptions of rainbow smelt, Arctic char and pink salmon revealed that all contribute to higher B-As concentrations. Subjects reporting to eat all four species among the anadromous species have nearly five times higher B-As than subjects reporting to eat none of them. No information is available on which As compounds are present in these fish species, and therefore, speciation analysis of As is warranted.

Humans are primarily exposed to Hg as methylmercury (MeHg), which is the predominant chemical form in marine and freshwater fish and other seafood, in particular, marine mammals and predatory fish. MeHg contributes with more than 90% of B-Hg concentrations in human populations consuming these foods [22,23]. Thus, we used total B-Hg amounts as a substitute measure of MeHg. Our results show that consumption of northern pike is one important source of Hg, which may explain the significantly higher B-Hg among the inland population that we have reported previously [12]. Fish and shellfish in the human diet contribute mostly to the B-Hg concentrations [24]. Since B-Hg is relatively high (4.0 μg/L) among non-consumers of northern pike, other fish species also contribute to the Hg dietary intake. This is in accordance with our earlier results showing that northern pike contains only around 3 to 4 times higher amounts of Hg than other local fish species [4]. In Arctic indigenous populations sea mammal meat consumption is an important source of exposure to Hg. None of the participants confirmed sea mammal consumption in their responses to the FFQ. However, an earlier study reports both hunting of sea mammals and consumption by inhabitants of Indiga, Nelmin Nos and Krasnoe [25]. Prohibitions for hunting sea mammals may have influenced the participants’ response in the FFQ.

There is an increasing understanding that not all MeHg in the diet is bioavailable. Some nutrients and food preparation methods appear to change the bio-accessibility. Studies of several marine fish species have shown overall mean absorption estimates ranging 12–79% [26]. Such information is not available for freshwater fish and especially for northern pike for which the molar ratio between Se and Hg is low. Selenium is reported to protect against toxic effects of Hg due to a high mutual binding affinity. How much Se is required to be protective against Hg toxicity is not known. There is some agreement that when the Se:Hg molar ratio is <1, Hg toxicity is likely to occur, and that a substantial excess of Se over Hg confers protection [27]. A molar Se:Hg ratio > 1 is often considered sufficient, but how much of an excess is really needed requires further, studies since not all Se in fish is chemically available to interact with Hg [27]. There are large significant differences in the molar Se:Hg ratios, ranging from 2.3 for northern pike up to 71.1 for pink salmon for fishes consumed among the Nenets population [4].

Selenium is abundant in animal meat and organs, seafood and fish, bread/cereals and dairy products, which are important nutritional sources for humans [28]. The B-Se concentrations among the participants were higher than needed for optimizing the function of important seleno-proteins [29]. Local meat and mushrooms were identified as possible sources of Se. Since most wild-grown mushrooms are poor in Se with amounts less than 1μg/g dry weight, it is not likely that the consumption of mushroom will contribute to the B-Se [30]. It was, however, expected that consumption of local meat would be identified as a source of Se. Self-reported consumption of fish was not associated with B-Se. However, in an alternative statistical approach, B-Hg and B-As were added to the statistical models. They are both regarded to be good markers of fish consumption [31]. They were both associated with B-Se, while mushroom and consumption of local meat were no longer associated with B-Se. This suggests that the total fish consumption contributes significantly to the Se status in this population. Further analysis showed that subjects in the lowest tertile of B-Hg and B-As had arithmetic mean B-Se concentration of 111 μg/L compared to 143 μg/L among those in the highest tertile of B-Hg and B-As. The Se wet weight content in different fish species consumed locally range from around 170 μg/kg in northern pike to 530 μg/kg in navaga [5]. We could not point out specific fish species that contribute significantly to B-Se. B-As is not applicable as a biomarker for fish consumption in areas with high arsenic concentrations in the drinking water, but to our knowledge, this is not the case in the NAO water bodies.

Since fish consumption is a major source of B-Hg among the participants, the speciation of Se in the local fish species may be of interest. A novel seleno-compound, selenoneine has been identified as one major form of Se in bluefish tuna and other marine organisms [32]. This Se analogue has been suggested to contribute to the scavenging of reactive oxygen species involved in methyl-Hg toxicity [33]. Inuits consuming Beluga skin rich in selenoneine exhibit a remarkably high B-Se with selenoneine as the major Se species in red blood cells (RBC) [34]. Since Se-methylselenoneine was only detected in RBC and not in the beluga skin, it is suggested that selenoneine undergoes methylation in humans, which may protect against methyl-Hg toxicity by increasing its demethylation in RBC and decreasing the distribution to target organs [35]. Further studies are needed to elucidate the contribution of selenoneine to the Se status of the indigenous populations of the Russian Arctic and whether it helps in the mitigation of potential adverse effects of methyl-Hg exposure.

Manganese is part of many human enzymes [36]. Whilst naturally occurring Mn deficiency has not been shown, Mn-induced neurotoxicity from respiratory and dietary exposures has been described [37]. Dietary intake of vegetable products is the major human source of Mn and it is interesting to note that there is a significant positive association between B-Mn and consumption of cloudberries [38]. The Mn content of cloudberries is, however, not well documented in the scientific literature.

Measurements of the Co content of reindeer meat and liver [39] found average amounts of 0.6 and 13.2 μg/100 g wet weight, respectively. Thus, dietary use of reindeer products may explain the slightly higher amounts of B-Co among those who have reported consumption of reindeer.

The increasing B-Pb with increasing number of meals with goose meat per month (Figure 5) is in line with earlier findings among indigenous Arctic populations and may be explained by Pb contamination of the meat by fragments of shotgun ammunition. Lead isotope ratio measurements of whole blood of First Nations people of Canada showed also that hunting activity (i.e., the use of a shotgun) was also an important source of Pb exposure [40]. Among male ethnic Greenlanders, it has been shown that consumption of hunted birds was the dominant association, with whole blood Pb concentrations pointing to lead shot as the major Pb source to people in Greenland [41].

Since the concentrations of Cu and Zn in plasma or serum are currently the best available biomarker of the risk of deficiency in a population, it is difficult to consider any insufficiency of these two essential elements using the measured whole blood concentrations measured in this study [42,43].

### 4.3. The Impact of Iron Status on Element Concentrations

B-Mn is negatively associated with S-ferritin, which is a marker of Fe status. B-Mn increases at S-ferritin concentrations <30–50 μg/L, which is much higher than what is recognized as Fe deficiency (ID). Additionally, B-Co increases at this S-ferritin level. Increases in B-Mn and B-Co in ID have been reported; however, the present study shows increased absorption also at lower, but sufficient Fe status. This may be caused by an upregulation of the duodenal Fe transporter divalent-metal transporter 1 [44], which is also known to transport Mn and Co. Higher B-Mn and B-Co have been measured in humans with ID, especially among healthy women of fertile age [15]. During pregnancy, the B-Mn is also enhanced [45]. Absorption of Mn among men has been reported to be lower than in women, which may be due to higher Fe stores among men [46]. When S-ferritin is accounted for, we are not able to show any significant gender difference in B-Mn, as also reported in a study of US residents [17].

### 4.4. The Impact of Lead on Element Concentrations

The B-Pb concentrations were positively associated with B-Mn, B-Hg, B-As, B-Zn and B-Cd in this study. These elements have in common that the highest amounts in the whole blood are found in the cellular blood fraction, largely consisting of erythrocytes. One mechanism of action of Pb on the whole blood concentrations may be related to its impairment of heme biosynthesis [47], although this has not been demonstrated in the literature. Lead affects the hematopoietic system by inhibiting the formation of haemoglobin by reducing key enzymes involved in the heme formation as well as ferro chelatase, which catalyses the incorporation of Fe into protoporphyrin [48]. The effects of Pb on the heme metabolism has been reported to start at around 0.1–0.3 μmol/L (20–60 μg/L) of B-Pb [48]. These concentrations are exceeded in many participants of this study, and it must therefore be assumed that many of them may be subject to slight effects on the hematopoietic system. The inhibition of ferro chelatase accumulates free erythrocyte protoporphyrin followed by substitution of Fe by Zn in the porphyrin ring to form Zn protoporphyrin [49]. In addition, in the case of ID or impaired Fe availability, Zn is an alternative substrate for ferro chelatase resulting in increased Zn protoporphyrin production [50].

Our observation of B-Mn increasing significantly with increasing B-Pb is compatible with a study of Pb-exposed children, where the increase in B-Mn with increasing B-Pb could be fully explained by the relation between free erythrocyte porphyrin and B-Mn [51]. Their conclusion was that the induced increased amounts of protoporphyrin in the erythrocytes takes up in vivo-available Mn-like Fe when not inhibited by Pb. In this process, Mn may also substitute with Zn, either as a Mn protoporphyrin or forming other Mn porphyrin compounds [52,53].

A significant positive relationship between B-Hg and B-Pb was observed earlier among Inuit women in Canada and was mainly explained by age [54]. This is, however, not confirmed in our study. We do not have any valid explanation for this difference.

Little is known about a potential impact of Pb on the blood concentrations of other elements. It is important to address this issue in future studies, because if a real impact exists, this may result in incorrect conclusions with respect to exposure associations.

## Figures and Tables

**Figure 1 nutrients-14-05005-f001:**
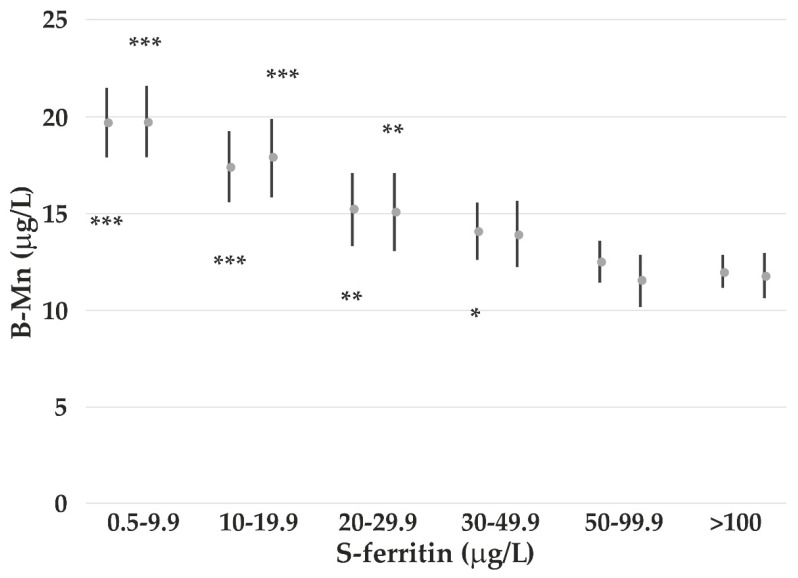
Arithmetic mean (and 95% CI) B-Mn concentrations among all participants (left) and females (right) adjusted for cloudberry consumption. The *p*-values (*** *p* < 0.001; ** *p* < 0.01; * *p* < 0.05) refer to the subgroup with S-ferritin >100 μg/L as reference for all participants and women, respectively.

**Figure 2 nutrients-14-05005-f002:**
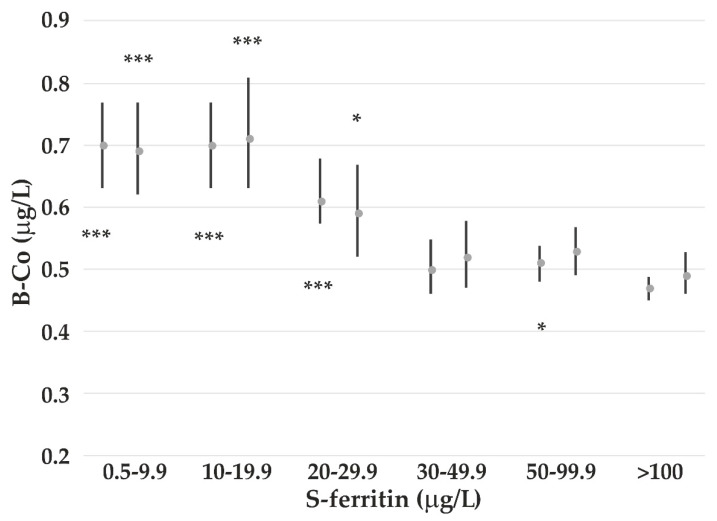
Geometric mean (and 95% CI) B-Co concentrations among all participants (left) and females (right) adjusted for age and reindeer consumption. The *p*-values (*** *p* < 0.001; * *p* < 0.05) refer to the subgroup with S-ferritin >100 μg/L as reference for all participants and women, respectively.

**Figure 3 nutrients-14-05005-f003:**
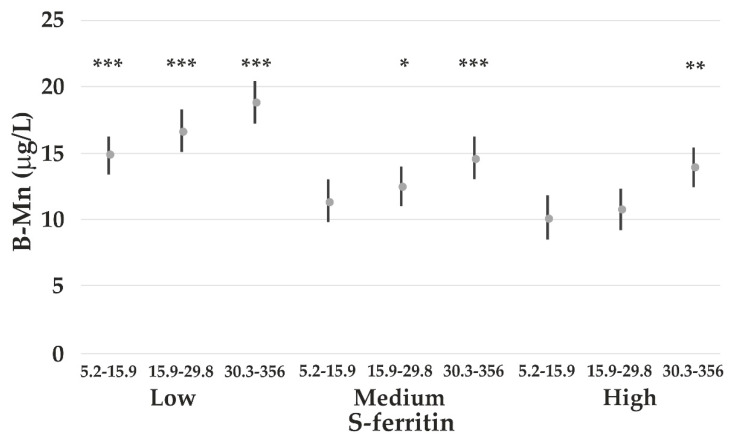
Arithmetic mean B-Mn concentrations among all participants according to S-ferritin divided into tertiles (low 0.5–44 μg/L; medium 45–108 μg/L; high 109–506 μg/L) and B-Pb concentrations (also divided into tertiles). The B-Pb concentration intervals are shown in the figure. The concentrations are adjusted for sex and cloudberry consumption. The *p*-values (*** *p* < 0.001; ** *p* < 0.01; * *p* < 0.05) refer to comparisons with the subgroup comprising high S-ferritin and low B-Pb.

**Figure 4 nutrients-14-05005-f004:**
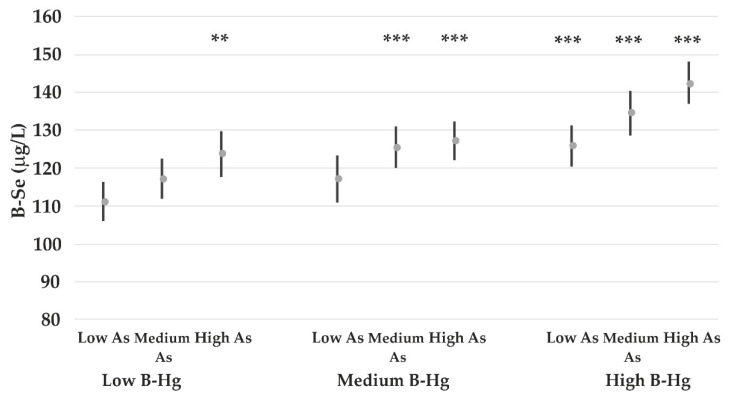
Arithmetic mean B-Se concentrations among all participants according to B-Hg stratified into tertiles (low 0.3–2.9 μg/L; medium 2.9–5.5 μg/L; high 5.5–24.3 μg/L) and B-As concentrations stratified into tertiles (low 0.5–3.5 μg/L; medium 3.5–8.8 μg/L; high 8.8–163 μg/L). The *p*-values (*** *p* < 0.001; ** *p* < 0.01) refer to comparisons with the subgroup comprising both low B-Hg and low B-As.

**Figure 5 nutrients-14-05005-f005:**
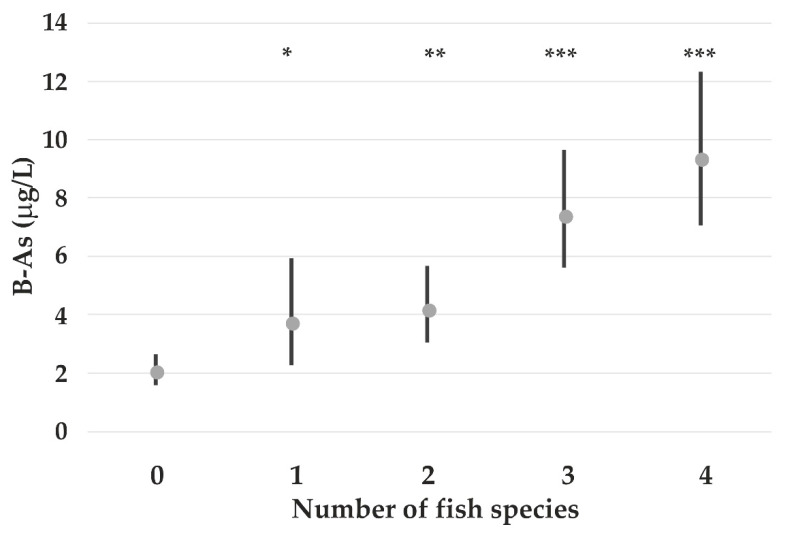
Geometric mean (and 95% CI) B-As concentrations among all participants according to number of selected fish species (navaga, rainbow smelt, pink salmon and Arctic char) they record to consume. The *p*-values (*** *p* < 0.001; ** *p* < 0.01; * *p* < 0.05) refer to comparisons with the subgroup reporting to consume none of the four species.

**Figure 6 nutrients-14-05005-f006:**
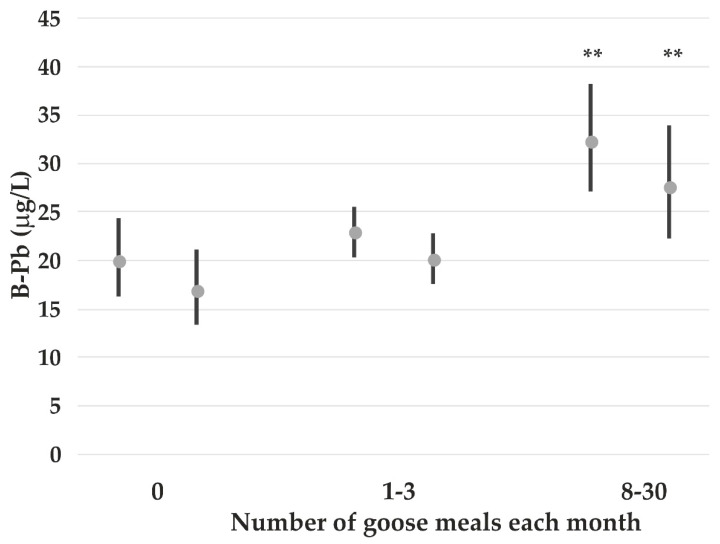
GM (and 95% CI) B-Pb concentrations among all participants (left) and non-hunters (right) according to number of meals with goose meat/month. The *p*-values (** *p* < 0.01) refer to comparisons between groups with no consumption of goose and the other respective subgroups.

**Table 1 nutrients-14-05005-t001:** The prevalence (in %) of subjects consuming local food items among female and male Nenets and non-Nenets inhabitants during a one-month period.

	Non-Nenets *n* = 136	Nenets *n* = 161	Women *n* = 204	Men *n* = 93	Coastal *n* = 148	Inland *n* = 149
	Prevalence (%)		Prevalence (%)		Prevalence (%)	
Fish						
Anadromous	88.9 *	78.9	84.2	81.7	88.4 *	78.5
Semi-anadromous	61.5 **	44.1	47.8 *	61.3	44.9 *	59.1
Marine	67.4	60.9	65.0	61.3	74.8 ***	53.0
Freshwater	59.3 *	46.6	52.2	52.7	29.7 ***	70.3
Meat Reindeer Other local meat (hare, elk, wild boar)	88.1 * 16.3	75.2 11.8	81.3 12.3	80.6 17.2	78.2 10.2	83.9 17.4
Birds Goose Other birds (duck and partridge)	81.5 52.6	82.5 43.5	82.2 44.3	81.7 54.8	88.4 ** 45.6	75.8 49.7
Berries and mushroom Mushroom Blueberries Cloudberries Cowberries Other berries	31.9 47.8 78.4 73.1 34.3 *	26.7 49.1 82.6 66.5 22.4	31.0 50.5 84.2 * 73.8 * 30.7	24.7 44.1 73.1 60.2 21.5	36.7 * 36.3 *** 77.4 58.9 *** 30.1	21.5 60.4 83.9 79.9 25.5
Whole blood concentrations ^c^	AM (min–max) μg/L	AM (min–max) μg/L	AM (min–max) μg/L	AM (min–max) μg/L	AM (min–max) μg/L	AM (min–max) μg/L
Co ^b^ Cu ^a^ Mn Se Zn ^a^ As ^b^ Cd ^b^ Hg Pb ^b^ S-ferritin	0.5 (0.3–1.7) 1.05 (0.58–1.9) ** 12.1 (4.4–27) *** 128 (89–227) ** 9.1 (4.2–13.7) ** 5.9 (0.5–143) ** 0.3 (<DL-4.2) ** 4.6 (0.5–18) ** 20 (5.4–356) *** 105 (3–506)	0.5 (0.3–2.5) 1.13 (0.73–2.4) 15.1 (4.3–35) 122 (85–192) 8.6 (4.1–14.5) 3.9 (0.5–163) 0.4 (<DL-3.0) 5.6 (0.3–24) 29 (5.2–281) 95 (0.5–506)	0.6 (0.3–2.5) *** 1.14 (0.69–2.4) *** 14.0 (4.3–35) 125 (85–227) *** 8.6 (4.2–14.5) 5.2 (0.5–163) 0.3 (<DL-4.2) 5.1 (0.3–24) 21 (5.2–356) *** 80 (0.5–463) ***	0.5 (0.3–0.9) 1.01 (0.58–1.5) 13.2 (4.5–24) 124 (89–192) 9.4 (4.1–12.9) 3.9 (0.5–143) 0.4 (<DL-2.6) 5.4 (0.6–24) 35 (5.4–281) 143 (13–506)	0.5 (0.3–2.5) 1.07 (0.58–2.4) ** 15.4 (5.3–35) *** 127 (85–192) 9.3 (4.9–13.7) *** 7.6 (0.5–163) *** 0.3 (<DL-4.2) 3.8 (0.3–21) *** 34 (7.2–356) *** 88 (0.5–463) *	0.5 (0.3–1.2) 1.13 (0.69–1.9) 12.0 (4.3–27) 123 (89–227) 8.4 (4.1–14.5) 3.0 (0.5–50) 0.3 (0.1–3.0) 6.5 (0.9–24) 17 (5.2–164) 111 (3–506)

* *p* < 0.05, ** *p* < 0.01, *** *p* < 0.001, AM = Arithmetic mean; ^a^ mg/L, ^b^ Geometric mean, ^c^ Sobolev et al., 2021 [12].

**Table 2 nutrients-14-05005-t002:** Pearson’s correlation coefficients calculated between essential and non-essential elements in whole blood among all participants.

	Co	Cu	Zn	Cd	Hg	Pb	As	Se
Mn	0.19 **	−0.20 **	0.22 **	0.11	−0.08	0.31 **	0.22 **	0.06
Co	-	0.26 **	−0.13 *	0.26 **	−0.17 **	−0.02	−0.09	−0.16 **
Cu		-	−0.33 **	0.05	0.06	−0.11*	−0.14 *	−0.11
Zn			-	0.14 *	0.02	0.30 **	0.10	0.23 **
Cd				-	0.02	0.25 **	−0.11	−0.07
Hg					-	0.15 **	0.06	0.33 **
Pb						-	0.15 **	0.16 **
As							-	0.32 **

* *p* < 0.05, ** *p* < 0.01.

**Table 3 nutrients-14-05005-t003:** Multiple linear regression analysis (backward procedure) including S-ferritin, gender, B-Pb (lg), current smoking, age, BMI, and various food items as independent variables.

	α				β			
		S-ferritin	Sex	B-Pb (lg)	Smoke	Age	BMI	Food item
B-Mn	6.8 ***	−0.02 ***	-	5.5 ***	-	-	-	1.8 Cloudberry **
B-Co (lg)	−0.20 ***	−0.0004 ***	−0.05 **	-	-	−0.001 *	-	0.04 Reindeer *
B-Se	97 ***	0.02 *	-	7.3 *	-	-	0.45 *	6.0 Mushroom * 7.3 Local meat *
B-As (lg)	−0.31 ^ns^	-	-	0.23 **	-	0.008 ***	-	0.44 Marine fish ***
B-Hg	0.18 ^ns^	0.009 ***	-	2.05 **	-	-	-	2.4 Freshw. Fish ***
B-Pb	1.20 ***	-	0.21 ***	♦	0.11 *	-	-	0.10 Goose *

Note: Sex: 1/0, 1 = men, 0 = women, Smoking: 1/0, 1 = current smoker, 0 = non-smoker, ♦: not included in this model; ^ns^ not significant; *** *p* < 0.001; ** *p* < 0.01; * *p* < 0.05.

## Data Availability

The data presented in this study are available on request from the corresponding author.

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
