# Peer review of "Diet and Blood Concentrations of Essential and Non-Essential Elements among Rural Residents in Arctic Russia"

_nutrients, 2022, doi:10.3390/nu14235005_

Round 1

Reviewer 1 Report

This article reveals that  the concentrations of B-Hg and B-As were highly associated with whole blood selenium, B-Mn increased significantly with increasing B-Pb, B-Pb were significantly associated with other  elements (Cd, Hg, Cu, Zn). It has great theoretical significance and high practical value for elements of intrinsic connections in the blood contribute to human health. The analysis process is comprehensive and the article is organized, smooth language and so on. Major revision can be published in Nutrients. However, there are some major issues need to be improved:

1. Abstract: The abstract should be modified to enhance the readability;

2. Introduction: The health effects of supplementary elements and the negative effects of heavy metals, the references try to correspond to the sentences quoted;

3. Results: It is necessary to explore the  healthy contribution of  the intrinsic connection of elements in the blood.

4. Discussion: Comparative analysis at home and abroad clarifies the innovation points of the paper

5. References: The strictly according to the format of Nutrients writing and supplement the latest references.

Author Response

  1. Abstract: The abstract should be modified to enhance the readability.

The abstract was re-written as suggested.

  1. Introduction: The health effects of supplementary elements and the negative effects of heavy metals, the references should correspond to the sentences quoted.

Thank you for spotting this inconsistency in referencing. We have corrected the references. Moreover, following the advice from another reviewer, we reduced the total number of references by 10%.

  1. Results: It is necessary to explore the healthy contribution of the intrinsic connection of elements in the blood

The paper already presents lots of results and exploration of health effects would further increase its size, which was already criticized by the second reviewer. Therefore, to keep the size of the paper within the reasonable limits and not to confuse the reader by several research hypotheses we kindly ask you to allow us to stick to the main aim of the paper without adding more information on health effects. They will be explored subsequent papers.

  1. Discussion: Comparative analysis at home and abroad clarifies the innovation points of the paper

Following the recommendation, we added a paragraph on comparisons of our findings with the results of other studies both in Russia and abroad

  1. References: The strictly according to the format of the Nutrients writing and supplement the latest references

We adjusted the format of the references to the style of the journal as requested. A few latest references were added as advised. At the same time, following the recommendations of the second reviewer, we reduced the total number of references by 10%.

Reviewer 2 Report

The authors aimed to describe the habitual intake (FFQ) and non-fasting serum response () of essential/non-essential elements (Co, Cu, Mn, Se, Zn, As, Cd, Hg, Pb + Ferritin) in Artic Asia residents (n= 297, 18-87 y, 69% female, 54% Nenets) from July-October 2018. Certain modifications are suggested to improve even more the scientific soundness and uniqueness of the manuscript: 

General

·         Sometimes the manuscript is hard to comprehend. The manuscript will improve even more if English grammar, syntax, and style are reviewed by a native English-spoken person or if the manuscript is reviewed by a formal translation agency.

·         There are too many unnecessary abbreviations (e.g. GM, ZPP are used only twice, FEP once), it is suggested to eliminate as many as possible. Some other are not defined (e.g. DL, line 171)

Title. It is somewhat long. please shorten.

Abstract. The authors should report this section in a more quantitative way and particularly highlight differential results from Nenets vs. others and by homeland region.

Introduction. Please shorten and highlight the absence (if any) of research on the studied area. Authors should define “Nenets” (line 68) and any other regionalism (along the manuscript) not commonly used by Nutrients ‘general audience

Methods. (2.2). Please provide more details as to the validation of the used FFQ. (2.3) Statement between lines 155-162 should be relocated in results section.

Results. Figures. They must be provided much higher resolution (300 dpi).

Discussion. The discussion should be more succinct, inductive>descriptive and in strict adherence to the results. There are several vague arguments not supported by evidence.

References. Too many references for an original research article. It is recommended to reduce to, say, 40 citations, eliminating multiple citations (e.g. statement between line 52) for the same statement or single citations (e.g. line 489, ref. 56). Also, check once again for unformatted or unproperly cited references (e.g. refs. 27, 29).

Author Response

General:

  • Sometimes the manuscript is hard to comprehend. The manuscript will improve even more if English grammar, syntax, and style are reviewed by a native English-spoken person, or the manuscript is reviewed by a formal translation agency.

The manuscript was revised for gramma, syntax, and style by our native English-speaking colleague in Norway.

  • There are too many unnecessary abbreviations (e.g. GM, ZPP are used only twice, FEP once)

All rarely used abbreviations were deleted.

  1. Title: It is somewhat long. Please, shorten

We slightly shortened the title as suggested. The new title is “Diet and blood concentrations of essential and non-essential elements among rural residents in Arctic Russia”. Further shortening is hardly possible without loss of important information.

  1. Abstract: the authors should report this section in a more quantitative way and particulary highlight differential results from Nenets vs. others and by homeland region

Thank you for this comment. We have thought about it earlier, but the text substantially exceeded the limits. The abstract was re-written following the recommendations from the first reviewer.  Inclusion of additional information with special emphasis on differential results from Nenets vs. others would have further complicated the reading. We hope that our revised version provides enough information on the main findings, but the readers will find the abovementioned comparisons in the main text of the paper.

  1. Introduction: please shorten and highlight the absence (if any) of research on the studied area. Authors should define Nenets (line 68) and any other regionalism (along the manuscript) not commonly used by Nutrients’ general readership.

We highlighted the scarcity of research on the studied area as suggested by the reviewer. Moreover, we added a definition of the Nenets ethnicity and the Nenets Autonomous Area in the introduction and along the manuscript as advised.

  1. Methods: Please, provide more details as to the validation of the used FFQ (2.3) Statement between lines 155-162 should be relocated in results section

Only face validity of the FFQ was assessed, which is a limitation of the paper. This limitation is described in the Discussion section. The statement between the lines 155-162 was relocated to the Results section as suggested by the reviewer.

  1. Results: Figures. They must be provided much higher resolution (300 dpi).

In the revised version we provide the figures with higher resolution as advised.

  1. Discussion: The discussion should be more succinct, inductive ->descriptive and in strict adherence to the results. There are several vague arguments not supported by evidence.

We managed to shorten the discussion. We also attempted to delete or revise all vague statements.

  1. References: Too many references for an original article. It is recommended to reduce to, say,. 40 citations, eliminating multiple citations (e.g. statement line 52) for the same statement or single citations (e.g. line 489, ref. 56). Also, check once again for unformatted or improperly cited references (e.g. refs. 27, 29)

We made all the changes mentioned by the reviewer. Moreover, we shortened the number of references as advised.

Round 2

Reviewer 2 Report

Thank you for addressing most of my observations, the manuscript has improved.